# GAMEBOT: Gaming Arena for Model Evaluation - Battle of Tactics

## Abstract

Large Language Models (LLMs) are increasingly deployed in real-world applications that demand complex reasoning. To track progress, we require robust benchmarks to evaluate their capabilities beyond superficial pattern recognition. However, existing benchmarks either suffer from data contamination or lack legibility. In this paper, we introduce GAMEBOT, a novel benchmark for evaluating LLMs in competitive gaming environments that addresses these limitations. GAMEBOT decomposes complex reasoning in games into modular subproblems, targeting abilities like rule understanding and strategy instruction following. We develop Chain-of-Thought (CoT) prompts that leverage domain knowledge to guide LLMs and automatically validate their intermediate reasoning steps against ground truth. This approach allows us to assess not only the accuracy of final decisions but also the quality of the underlying reasoning process. We benchmark 17 prominent LLMs across eight diverse games, encompassing various strategic abilities and game characteristics. GAMEBOT offers four advantages: (1) Mitigation of Data Contamination: Dynamic game states minimize overlap with pre-training data. (2) Legibility: Evaluation of intermediate reasoning steps enables fine-grained scrutiny of LLM behavior. (3) Difficulty: The games effectively differentiate top-performing models. (4) Stronger Baselines: Our curated CoT prompts establish competitive baselines for future research. We hope GAME-BOT stimulates further work that seeks a deeper understanding of LLM reasoning capabilities in strategic settings.

## 1 Introduction

While large language models (LLMs) have demonstrated impressive capabilities across a variety of tasks like translation, question answering and coding (Achiam et al., 2023; Reid et al., 2024; Anthropic, 2024a), their increasing integration into commercial products necessitates robust benchmarks for evaluating their reasoning abilities. Existing efforts have focused on creating benchmarks that move beyond superficial pattern recognition and delve into the core reasoning skills required for problem-solving. For instance, GSM8K (Cobbe et al., 2021) and MATH (Hendrycks et al., 2021) target mathematical reasoning, HumanEval (Chen et al., 2021) and MBPP (Austin et al., 2021) assess code generation abilities, and StrategyQA (Geva et al., 2021) focuses on multi-hop reasoning.

However, the pre-training of LLMs relies on extensive web-scale corpora, which inadvertently encompass instances from evaluation benchmarks. This leads to the phenomenon of *data contamination*, where LLMs memorize test instances rather than exhibiting reasoning capabilities (Yang et al., 2023; Xu et al., 2024; Zhang et al., 2024). Data contamination has sparked significant concern within the research community, as it poses a substantial threat to the reliability of LLM evaluations.

To mitigate the impact of data contamination, several avenues have been explored, including human annotation-based methods (White et al., 2024; Chiang et al., 2024; Kiela et al., 2021; Zheng et al., 2023) or LLM-driven approaches (Sprague et al., 2023; Fu et al., 2024; Dubois et al., 2024; Zeng et al., 2023). These methods can be either costly or susceptible to biases inherent in the employed LLMs (Panickssery et al., 2024).

Differently from these two methods, recent research has leveraged strategic gaming as a testbed, which naturally contains complex and dynamic environments with well-defined rules and objectives (Liu et al., 2023; Huang et al., 2024; Duan et al., 2024; Chalamalasetti et al., 2023; Chen et al.,

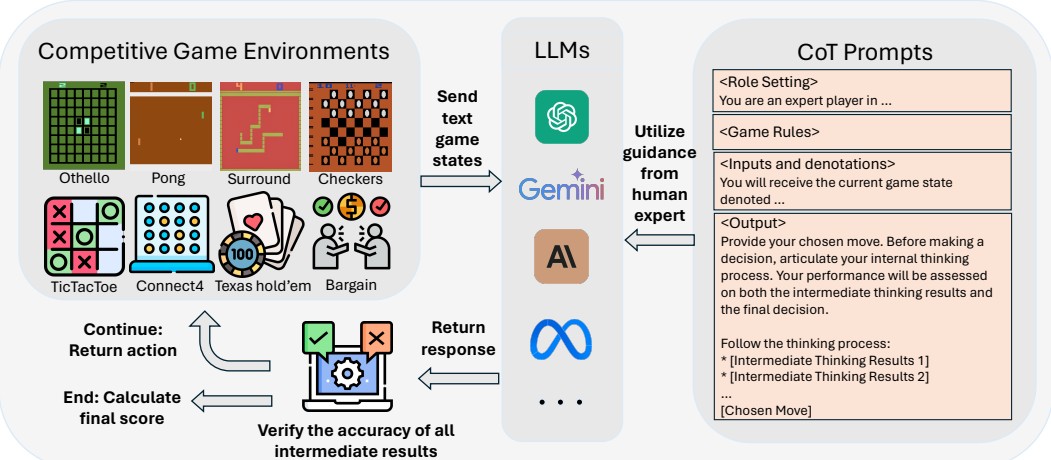

Figure 1: Overall evaluation framework of GAMEBOT.

2024; Wu et al., 2023). This enables a quantitative evaluation of LLM performance and pushes LLMs to exhibit reasoning at the limits of their capabilities. However, existing studies only rely on game outcomes, employing metrics such as win rate or score, failing to capture the nuances of the decision-making process that underlies these results. For instance, an LLM might produce non-sensical reasoning yet select the correct action, leading to a fortuitous victory. This undermines the reliability of evaluation. Besides, analyzing only the final outcome of a game, which may comprise hundreds of steps, sacrifices the rich information embedded within the LLM's decision-making process. To comprehensively assess the capabilities of LLMs in strategic settings, it is crucial to evaluate not only the final outcomes but also the intermediate thoughts underpinning LLM actions. This deeper level of analysis can reveal valuable insights into the strengths and weaknesses of different LLMs, paving the way for more targeted improvements in their design and training.

In this paper, we introduce **GAMEBOT**, a benchmark for evaluating LLMs in competitive gaming environments. Specifically, we decompose complex reasoning problems in games into a few logi-cally essential subproblems, designed to evaluate LLMs' abilities in rule understanding, game state comprehension, and adherence to strategic instructions. Each modular subproblem contributes to the final decision, and can be automatically validated against ground truth generated by programs. Instead of using generic "think step by step" prompts, we develop Chain-of-Thought (CoT) prompts that provide detailed strategic guidance for LLMs. During competitions in dynamic games, LLMs are instructed to demonstrate their intermediate reasoning steps (results of subproblems) alongside their final action within a unified response. This approach evaluates the consistency of LLMs' rea-soning across complex multi-hop scenarios. We show that the decomposition of problems not only significantly improves the performance of LLMs, but more importantly, also enable us to go beyond only measuring win/loss outcomes, by assessing the accuracy of intermediate reasoning steps to-gether. In this way, our proposed benchmark provides valuable insights into specific areas where LLMs excel or struggle, giving the final evaluation results greater legibility.

GAMEBOT comprises eight challenging games spanning a diverse taxonomy, including board games, action games, card games, and game-theoretic games. These games target distinct strate-gic abilities and encompass various game characteristics: *zero-sum* (e.g., Othello, Checkers, Tic-Tac-Toe, Connect4) versus *non-zero-sum* (e.g., Bargaining); *perfect information* versus *imperfect information* (e.g., Texas Hold'em); and *turn-based* versus *simultaneous move* games (e.g., Pong, Surround). These tasks and our designed subproblems require a variety of abilities, such as spatial reasoning, collaboration in competition, math equation solving, long-context information extraction, risk management and pattern recognition.

We benchmarked LLMs through one-versus-one direct competition, augmented by a random player as a baseline. The games are evolving dynamically based on the actions from both players, ensuring a diverse range of game states. We evaluate 17 prominent LLMs, including GPT, Claude, Gemini, LLama, Mistral, and others. The final results are shown in Figure 2. We make some key obser-vations: (1) Model size demonstrably affects performance in our challenging, reasoning-intensive

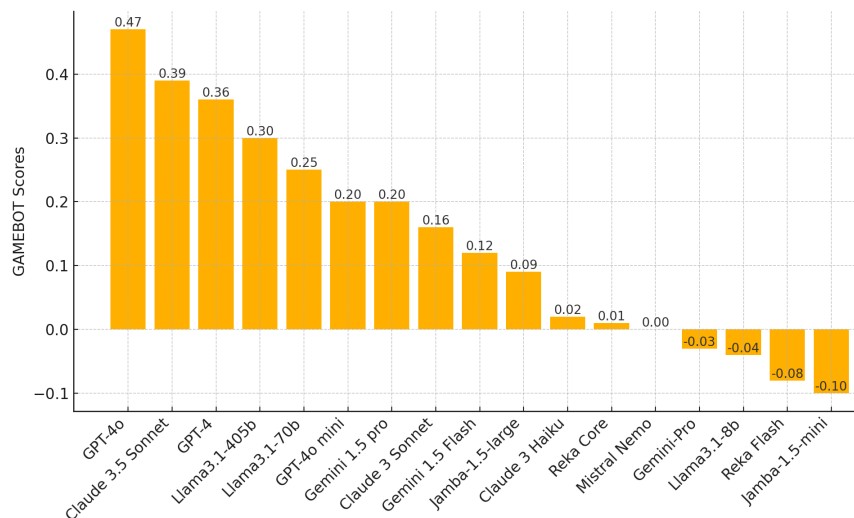

Figure 2: Overall LLM ranking and performance in GAMEBOT.

benchmarks. Among them, the performance of GPT-4o mini significantly lags behind GPT-4o, and even slightly worse than GPT4. This finding contrasts with the trend observed in the Chatbot Arena Leaderboard (Chiang et al., 2024). (2) The verification of intermediate results in LLMs is highly predictive of final game outcome performance, suggesting the robustness of our benchmark. (3) GPT-4o and Claude 3.5 Sonnet exhibit consistently strong performance, showing more capabilities in generalization and complex reasoning.

To summarize, our GAMEBOT offers the following advantages. **Mitigation of Data Contamination:** Rather than evaluation on a predefined dataset, we evaluate LLMs in interactive gaming environments, whereas game states are across a wide spectrum depending on specific actions received and randomness. **Legibility:** Our benchmark offers assessments on not only the quality of final decisions but also the intermediate thoughts, giving insights for improving the training or inference of LLMs. **Difficulty:** The games are challenging enough to differentiate between top-performing models. **Stronger Baselines:** Our curated prompts also serve as much stronger CoT baselines than previous methods (Duan et al., 2024; Chen et al., 2024). The prompts presented in this work can serve as valuable CoT baselines for future research exploring advanced prompting techniques like auto-prompting (Zhang et al., 2022) and reflection (Shinn et al., 2024).

## 2 GAMEBOT

In this section, we detail the experimental setup and results of evaluating LLMs within game environments, and provide analysis of the performance of various LLMs.

### 2.1 BENCHMARK CONSTRUCTION

GAMEBOT provides a comprehensive suite for analyzing LLMs' reasoning ability, including developed gaming environments, human-crafted prompts assisted by experts with domain gaming knowledge, and programs for assessing the accuracy of intermediate thoughts.

We implemented custom environments for Checkers, Bargaining, Othello, and Tic-Tac-Toe, providing standardized interfaces for evaluation. For Pong, Surround, Texas Hold'em, and Connect4, we leveraged the PettingZoo multi-agent environment framework (Terry et al., 2021). While Pong and Surround inherently provide only pixel-based visual information, we extracted relevant representations and translated them into textual form following Anand et al. (2019), maintaining a consistent text-based game state representation across all environments.

To validate the LLM intermediate thoughts, we also develop programs to automatically generate ground truth answers for each game's subproblems, enabling efficient and objective evaluation.

Table 1: Games for evaluation.

| Games | Game Properties | | | | Representative Evaluated Abilites |
|-------|------|-------------|--------------|---------|---------------------------------|
|       | Type | Information | Simultaneous | Zero-sum | |
| Othello | Board Game | Perfect | No | Yes | Spatial Reasoning |
| Pong | Action Game | Perfect | Yes | Yes | Mathematical Reasoning |
| Surround | Action Game | Imperfect | Yes | Yes | Long-Context Information Extraction |
| Checkers | Board Game | Perfect | No | Yes | Spatial Reasoning |
| TicTacToe | Board Game | Perfect | No | Yes | Pattern Recognition |
| Connect4 | Board Game | Perfect | No | Yes | Pattern Recognition |
| Texas hold'em | Card Game | Imperfect | No | Yes | Risk Management |
| Bargaining | Game Theoretic | Imperfect | No | No | Collaboration in Competition |

## 2.2 EVALUATION IN COMPETITION

We evaluate LLMs through their direct competition in dynamic game environments, rather than via traditional single-agent evaluation. During pre-training of LLMs, while some game states might be inadvertently memorized, the exponential state space growth inherent in dynamic games renders exhaustive memorization impractical. However, single-agent benchmarks, often employing fixed-policy opponents, explore only a limited subset of the potential game states, increasing vulnerability to data contamination. Our competitive framework, by pitting LLMs against adaptive adversaries, forces them to navigate a vastly more diverse and unpredictable landscape of game states, providing a more robust test of their strategic reasoning abilities and mitigating the impact of potential data contamination.

## 2.3 INCLUDED GAMES AND SUBPROBLEM DECOMPOSITION

GAMEBOT comprises eight diverse games carefully selected to encompass a wide range of strategic abilities and game characteristics (See Table 1). This diverse selection allows us to evaluate LLMs across different reasoning dimensions, such as spatial reasoning, opponent modeling, risk management, and collaboration. To facilitate a fine-grained analysis of LLM reasoning, we decompose the complex decision-making process within each game into 2-3 logically essential subproblems. Each subproblem targets a specific aspect of the game's reasoning requirements and contributes to the final action selection.

This decomposition offers three key advantages: **(1) Finer-Grained Evaluation** – Existing benchmarks for evaluating LLMs in gaming scenarios (Wu et al., 2023; Duan et al., 2024; Chen et al., 2024; Liu et al., 2023) typically rely solely on final game outcomes (e.g., win or lose) as the evaluation metric. However, a single game can involve numerous steps and be influenced by numerous chance occurrences. It is possible for an LLM to exhibit reasonable gameplay throughout most of a match but ultimately lose due to a single critical misstep. Consequently, relying solely on final outcomes can lead to unreliable and unstable evaluations, lacking robustness. In contrast, our framework evaluates the quality of reasoning at each step by automatically verifying the LLM's answers to subproblems, thereby providing a more comprehensive assessment of the entire gaming process. **(2) Insights into Specific Strengths and Weaknesses** – Our designed subproblems for each game target specific cognitive abilities, such as rule understanding and adherence to strategic instructions. By analyzing LLM performance on each subproblem, we can potentially reveal the capabilities and weaknesses of the evaluated models in these targeted areas. **(3) Enhanced Strategic Decision-Making** – The predefined subproblems contribute to the strategic decision-making process. By explicitly guiding LLMs to address each subproblem before making the final decision, we encourage a more structured and deliberate approach to gameplay. This structured reasoning process improve the overall reasoning abilities and strategic planning of the LLMs. (See Figure 3 as an example.)

We now detail the included games and their respective subproblem designs.

**Othello (Reversi)** Othello is a board game played on an 8x8 board. Two players take turns placing discs of their color, attempting to outflank and capture their opponent's discs by sandwiching them between their own. The captured discs would be flipped to the player's color. The object of the

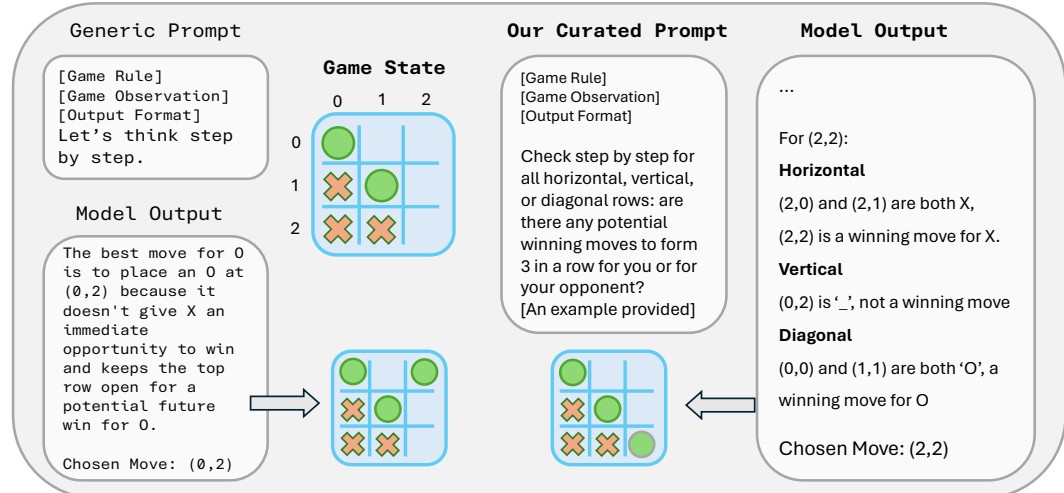

Figure 3: Impact of curated prompts on Llama 3.1 70B performance in TicTacToe. The left panel shows the model's output with a basic prompt, while the right panel demonstrates the improved output achieved with our curated prompt. Outputs are from actual model generations.

game is to have the majority of pieces showing the player's color at the end of the game. The game emphasizes strategic placement and tactical maneuvering to control the board.

**Subproblem Design:** 1. *Output whether you have a move to directly occupy the corners.* 2. *A 'wedge' in Othello is when a player can place a piece between two of the opponent's stable pieces on the edge, ..., output all of the coordinates that can create a wedge.*

Evaluated abilities: Spatial Reasoning; Positional Evaluation

**Pong**  Pong is a classic two-player arcade game simulating table tennis. Players control paddles to hit a ball back and forth, aiming to score points by making the opponent miss. It represents a simplified environment with continuous action spaces.

**Subproblem Design:** 1. *Output the moving direction of the ball.* 2. *Output the y-coordinate of the ball when its x-coordinate is the same as your paddle's x-coordinate.*

Evaluated abilities: Mathematical Reasoning

**Surround (Snake)**  Surround is a two-player game where players control a continuously moving line. The goal is to force the opponent to collide with their own line, a wall, or the growing trail of either player. It highlights spatial reasoning and strategic blocking.

**Subproblem Design:** 1. *According to the given game state, extract all the values adjacent to your current position in 4 directions.* 2. *List all possible move actions based on the available empty spaces around your current position.* 3. *Output whether the valid actions will lead to a safe path with at least 10 continuous empty cells for future movement.*

Evaluated abilities: Long-Context Information Extraction; Long-Term Planning

**Checkers (Draughts)**  Checkers is a board game where players move their pieces diagonally, capturing opponent pieces by jumping over them. Regular pieces can only move forward, while "kings," earned by reaching the opponent's back rank, can move and capture both forwards and backward. The game ends when one player has captured all of their opponent's pieces or has blocked their opponent's pieces, leaving them with no legal moves. It involves strategic planning and tactical piece advancement, with the objective of capturing all opponent pieces or blocking their movement.

**Subproblem Design:** 1. *Output all of the moves that give you a new king piece.* 2. *Output all of the bad moves that lead to a worthless die.* 3. *Two-for-One Shot is ..., output all of the moves that can create a Two-for-One Shot.*

Evaluated abilities: Spatial Reasoning; Game Board Understanding

**Tic-Tac-Toe (Noughts and Crosses)**   Tic-Tac-Toe is a simple two-player game played on a 3x3 grid. Players take turns marking a square with their respective symbol, aiming to create a line of three symbols horizontally, vertically, or diagonally. Its simplicity makes it useful for a lightweight evaluation for LLMs. However, note that its game states are limited compared to other games included in the benchmark.[1] Nevertheless, we found it remains challenging for LLMs.

**Subproblem Design:** 1. *Are there any potential winning moves to form 3 in a row for you? 2. Are there any potential winning moves to form 3 in a row for your opponent?*

Evaluated abilities: Pattern Recognition; Game Board Understanding

**Connect Four**   Connect Four is a two-player connection game played on a vertically suspended 6x7 grid. Players drop colored discs into columns, aiming to connect four of their own discs horizontally, vertically, or diagonally. It involves strategic thinking and anticipating opponent moves.

**Subproblem Design:** 1. *Are there any potential winning moves to form 4 in a row for you? 2. Are there any potential winning moves to form 4 in a row for your opponent?*

Evaluated abilities: Pattern Recognition; Game Board Understanding

**Texas Hold'em**   Texas Hold'em is a popular variant of poker involving betting, bluffing, and incomplete information. Players receive two private cards and share five community cards, forming the best possible five-card hand. Multiple betting rounds occur throughout the hand, allowing players to bet strategically based on the strength of their hand and their assessment of their opponents' hands. The player with the best hand at the showdown, or the last remaining player after all others have folded, wins the pot. It presents a challenging environment with imperfect information and complex strategic considerations.

**Subproblem Design:** 1. *The winning probabilities of given private hand are ..., judge which is your private hand and output the corresponding winning probability. 2. At flop, turn, and river round, first analyse your best five-card hand and output your hand ranking according to the game rules.*

Evaluated abilities: Risk Management; Bluffing; Hand analysis

**Bargaining**   Bargaining (Lewis et al., 2017) is a game where two players negotiate to divide a set of items, each holding a private valuation for each item. To ensure diverse game states and richer strategic interactions, we modify the standard setting by increasing the total value of the items to 30 for each player. Players negotiate to maximize their individual total value acquired. Furthermore, we introduce a dynamic setting: after 8 rounds of bargaining, the game has a 20% chance of ending in each subsequent round. If no agreement is reached before the game's forced termination, both players receive a reward of 0. This modification incentivizes players to consider both individual gain and collaborative outcomes. Bargaining games explore concepts of cooperation, competition, and fairness in resource allocation.

**Subproblem Design:** 1. *Based on the previous rounds of bargaining, evaluate the opponent's latest proposal and calculate the total value of the items for you and output the result. 2. For your own valid proposal, output the total value of the items for you.*

Evaluated abilities: Collaboration in Competition; Opponent Modeling; Mathematical Reasoning

## 2.4 EVALUATION METRICS

**Outcome Evaluation**   Agent performance is evaluated using two distinct metrics based on game outcome structure. For win/draw/lose outcomes (Othello, Pong, Surround, Checkers, TicTacToe, Connect4), the metric is: $s_1 = (W - L)/N$, where $W$, $L$, and $N$ denote the number of wins, losses, and matches played, respectively. For value-based outcomes (Texas Hold'em, Bargaining), the metric is: $s_2 = (\sum V_{\text{win}} - \sum V_{\text{lose}})/N$, $s_2$ is then normalized to a maximum of 0.5: $s_2' = s_2/\alpha$,

---

[1] The total number of possible legal game states for Tic-Tac-Toe is 5,478 (Wikipedia).

Table 2: LLM performance based on final game outcomes. (For the Checkers benchmark, models marked with "-" rarely generated valid moves. To facilitate the calculation of average scores, these cases were treated as equivalent to a score of -0.5.)

| LLMs | Othello | Pong | Surround | Checkers | TicTacToe | Connect4 | Texas hold'em | Bargaining | Average |
|---|---|---|---|---|---|---|---|---|---|
| GPT-4o | **0.35** | **0.45** | 0.62 | **0.27** | 0.34 | **0.45** | **0.50** | 0.35 | **0.42** |
| GPT-4o mini | -0.36 | 0.07 | 0.33 | -0.19 | 0.05 | -0.14 | 0.16 | 0.42 | 0.05 |
| GPT-4 | 0.12 | 0.06 | 0.59 | 0.01 | 0.26 | 0.36 | 0.47 | 0.33 | 0.28 |
| Gemini 1.5 pro | 0.19 | 0.29 | -0.34 | 0.15 | -0.05 | -0.16 | 0.14 | 0.12 | 0.05 |
| Gemini 1.5 Flash | -0.13 | -0.03 | 0.45 | 0.05 | -0.31 | 0.04 | -0.23 | 0.09 | -0.01 |
| Gemini-Pro | -0.10 | -0.28 | -0.30 | - | -0.11 | -0.40 | 0.40 | -0.15 | -0.17 |
| Claude 3.5 Sonnet | 0.31 | 0.12 | **0.63** | -0.07 | 0.21 | 0.35 | 0.37 | **0.50** | 0.30 |
| Claude 3 Sonnet | 0.07 | 0.37 | -0.49 | -0.05 | 0.15 | 0.27 | 0.20 | -0.16 | 0.05 |
| Claude 3 Haiku | 0.07 | -0.32 | -0.14 | -0.41 | -0.11 | -0.34 | 0.04 | -0.07 | -0.16 |
| Reka Core | -0.11 | -0.15 | -0.45 | - | -0.12 | 0.23 | 0.12 | -0.28 | -0.16 |
| Reka Flash | -0.35 | -0.25 | -0.38 | - | -0.27 | -0.15 | -0.45 | -0.09 | -0.31 |
| Llama3.1-405b | 0.11 | 0.33 | 0.61 | -0.12 | 0.17 | 0.36 | -0.19 | 0.10 | 0.17 |
| Llama3.1-70b | 0.20 | 0.26 | 0.14 | -0.06 | **0.47** | 0.26 | -0.23 | 0.03 | 0.14 |
| Llama3.1-8b | -0.13 | -0.29 | -0.44 | - | -0.05 | -0.11 | -0.33 | -0.27 | -0.26 |
| Jamba-1.5-large | 0.07 | -0.20 | -0.14 | 0.18 | 0.01 | 0.04 | 0.12 | 0.15 | 0.03 |
| Jamba-1.5-mini | -0.01 | -0.38 | -0.31 | - | -0.29 | -0.38 | -0.30 | -0.21 | -0.30 |
| Mistral Nemo | -0.02 | -0.20 | -0.54 | -0.08 | -0.16 | -0.21 | 0.35 | -0.20 | -0.14 |
| *Random* | -0.27 | -0.53 | -0.44 | -0.01 | -0.39 | -0.49 | -0.56 | -0.47 | -0.40 |

where $\alpha$ is a scaling constant ensuring $\max(s_2') = 0.5$. We thus obtain the final outcome evaluation, denoted by $S$, by averaging $s$ across eight games.

**Intermediate Thought Evaluation**  LLM intermediate thought performance is evaluated per sub-problem using either F1 score or accuracy. The F1 score is employed for problems with unbalanced answer distributions, providing a more robust evaluation in such cases. For problems with balanced answer distributions, accuracy is used. The overall performance metric, denoted by I, is computed as the average of the individual subproblem results across the entire game: $I = \frac{1}{t} \sum_{i=1}^{t} M_i$, where $M_i$ is the result for each subproblem.

**Final Metrics**  The final metric is the average of S and I, Scores $= (S + I)/2$.

## 3 EXPERIMENTS

### 3.1 GAMEBOT BENCHMARKING

In this section, we introduce the evaluated LLMs and settings for the whole benchmark. We also reveal some key observations on the experimental results.

#### 3.1.1 EVALUATED LLMS

We benchmark 17 LLMs on our GAMEBOT. Where possible, we focus on chat or instruction-tuned variants as they typically have stronger instruction-following abilities. We include the following LLMs in our evaluation:

**Closed-source**: GPT-4 (Achiam et al., 2023), GPT-4o (OpenAI, 2024a), GPT-4o mini (OpenAI, 2024b), Gemini 1.5 Pro, Gemini 1.5 Flash (Reid et al., 2024), Gemini-Pro (Gemini Team et al., 2023), Claude 3 Haiku, Claude 3 Sonnet (Anthropic, 2024a), Claude 3.5 Sonnet (Anthropic, 2024b), Reka Core and Reka Flash (Ormazabal et al., 2024).
**Open-source**: LLaMA 3.1 (8B, 70B, 405B) (Dubey et al., 2024), Jamba 1.5 (Large, Mini) (Team et al., 2024), and Mistral Nemo (AI, 2024a).

We include details of the specific model versions used in the Appendix A.

#### 3.1.2 SETTINGS

We carry out inference using the default sampling parameters of each LLM. By using the default parameters, we ensure non-deterministic output, introducing more diversity. This allows us to carry

Table 3: LLM performance based on intermediate result verification.

| LLMs | Othello | Pong | Surround | Checkers | TicTacToe | Connect4 | Texas hold'em | Bargaining | Average |
|---|---|---|---|---|---|---|---|---|---|
| GPT-4o | **0.44** | 0.92 | 0.43 | **0.27** | **0.61** | 0.18 | **0.85** | 0.44 | **0.52** |
| GPT-4o mini | 0.01 | 0.79 | 0.34 | 0.16 | 0.29 | 0.05 | 0.63 | **0.57** | 0.36 |
| GPT-4 | 0.15 | 0.89 | 0.50 | 0.17 | 0.55 | **0.19** | 0.55 | 0.43 | 0.43 |
| Gemini 1.5 pro | 0.20 | 0.88 | 0.22 | 0.25 | 0.18 | 0.07 | 0.63 | 0.25 | 0.34 |
| Gemini 1.5 Flash | 0.01 | 0.96 | 0.48 | 0.09 | 0.07 | 0.05 | 0.26 | 0.14 | 0.26 |
| Gemini-Pro | 0.08 | 0.51 | 0.04 | 0.00 | 0.05 | 0.01 | 0.10 | 0.05 | 0.11 |
| Claude 3.5 Sonnet | 0.25 | **0.97** | **0.61** | 0.17 | 0.58 | 0.09 | 0.70 | 0.45 | 0.48 |
| Claude 3 Sonnet | 0.13 | 0.92 | 0.21 | 0.07 | 0.18 | 0.01 | 0.41 | 0.18 | 0.26 |
| Claude 3 Haiku | 0.09 | 0.80 | 0.25 | 0.05 | 0.01 | 0.00 | 0.27 | 0.12 | 0.20 |
| Reka Core | 0.02 | 0.80 | 0.05 | 0.00 | 0.03 | 0.04 | 0.16 | 0.27 | 0.17 |
| Reka Flash | 0.00 | 0.70 | 0.04 | 0.00 | 0.04 | 0.03 | 0.31 | 0.07 | 0.15 |
| Llama3.1-405b | 0.32 | 0.95 | 0.43 | 0.12 | 0.48 | 0.16 | 0.68 | 0.41 | 0.44 |
| Llama3.1-70b | 0.07 | 0.89 | 0.46 | 0.16 | 0.52 | 0.09 | 0.47 | 0.23 | 0.36 |
| Llama3.1-8b | 0.15 | 0.77 | 0.04 | 0.00 | 0.10 | 0.02 | 0.20 | 0.07 | 0.17 |
| Jamba-1.5-large | 0.11 | 0.53 | 0.21 | 0.05 | 0.16 | 0.00 | 0.07 | 0.09 | 0.15 |
| Jamba-1.5-mini | 0.14 | 0.52 | 0.02 | 0.00 | 0.06 | 0.02 | 0.02 | 0.05 | 0.10 |
| Mistral Nemo | 0.19 | 0.59 | 0.03 | 0.03 | 0.05 | 0.00 | 0.13 | 0.12 | 0.14 |

out repeat LLM head-to-head competitions in which the models are exposed to novel game states and positions, resulting in a more comprehensive evaluation of ability. For each LLM, we set the maximum number of output tokens parameter to 4096 to allow sufficient tokens for reasoning steps.

In each game environment, we conduct 20 matches between each pair of models, with each LLM playing 10 matches as the first player and 10 as the second to mitigate first-player advantage. We also set a "*Random Player*" which randomly choose an available move as a baseline for better interpretation of the results.

In the prompts, we meticulously detail game rules to make them self-contained, and standardize the format of inputs and expected outputs from LLMs. The set of prompts is not specifically optimized for any individual LLM, ensuring fairness. The whole prompt suite can be found in Appendix B.

### 3.1.3 RESULTS

LLM performance is evaluated based on final game outcomes (Table 2) and intermediate result verification (Table 3). These results reveal the following key observations:

**Observation 1: Impact of Model Size** Model size demonstrably affects performance in our challenging, reasoning-intensive benchmarks. Larger models consistently outperformed smaller models within each series, demonstrating the importance of capacity for these challenging tasks. For lightweight models like Reka Flash and Jamba-1.5-mini, they exhibited performance nearing random levels. Surprisingly, a substantial performance gap was observed between GPT-4o and GPT-4o mini. Besides, despite being an older version, GPT-4 still outperformed GPT-4o mini, contrasting with the trend observed in the Chatbot Arena Leaderboard (Chiang et al., 2024). All these findings underscore the importance of model scale in our sophisticated reasoning task.

**Observation 2: Correlation Between Final Outcomes and Intermediate Steps** Looking into the overall performance, the verification of intermediate results in LLMs is highly predictive of final game outcome performance. For instance, models struggling with intermediate steps, such as Gemini-Pro, Reka Flash, and Jamba-1.5-mini, also performed poorly in terms of final outcomes according to the average results. This finding highlights the crucial role of intermediate step verification in understanding and evaluating LLM performance. This verification provides a window into the LLM's decision-making, giving clues to the "why" behind its actions and making the final outcomes less opaque[2].

However, a closer examination of individual game performance reveals some exceptions to this general trend. For example, while Claude 3.5 Sonnet achieves the highest score in intermediate step verification for Pong (Table 3), its corresponding final outcome score is not as impressive. This sug-

---

[2]We note, however, the limitations of this form of interpretability (the externalized reasoning may not fully reflect the underlying decision process (Turpin et al., 2024)).

gests that while accurate assessment of intermediate states is generally a strong indicator of success, other factors can also influence the final outcome, potentially including game-specific strategies, risk tolerance, or even chance elements within certain games. We analyze this phenomenon in more detail in the following subsection.

**Observation 3: Inconsistency Across Games**   Many models exhibit unstable performance across different games. For example, Llama3.1-70b achieves the highest final outcome score in TicTacToe unexpectedly and performs relatively well in Pong and Connect4, yet in Texas hold'em, it obtains a negative score. This variability suggests that, like some other models, Llama3.1-70b may be learning game-specific strategies rather than developing generalizable reasoning abilities. These performance fluctuations highlight the challenge of developing LLMs capable of robust and consistent decision-making across diverse scenarios, potentially indicating limitations in their ability to transfer knowledge and adapt to new game rules.

**Observation 4: Strength of GPT-4o and Claude 3.5 Sonnet**   Both GPT-4o and Claude 3.5 Sonnet exhibit consistently strong performance across both evaluation metrics, achieving the highest average scores in both tables. This suggests that these models possess more robust game-playing capabilities compared to other LLMs considered.

### 3.1.4 Deeper Investigation into the Results

We further investigate the underlying reasons behind some of the unexpected performance patterns observed.

**Claude 3.5 Sonnet's performance on Pong**   As previously noted, Claude 3.5 Sonnet's strong intermediate performance in Pong does not translate to a similarly high final outcome score. Manual review of video and log files revealed the cause: while the model accurately predicted the ball's position and positioned its paddle accordingly, it rigidly adhered to centering the paddle on the ball. This ignored the instruction to intercept using the paddle's corner. The game's frame-skipping mechanism sometimes rendered precise centering impossible, leading to paddle oscillation near the target and occasional missed balls.

**Llama3.1-70b's performance on Texas Hold'em**   While Table 2 indicates that Llama3.1-70b underperforms in Texas Hold'em, a closer examination of the intermediate results in Table 3 suggests its reasoning abilities are stronger than the final performance might imply. We observed that the game's high-risk nature contributes to this discrepancy. Specifically, when Llama3.1-70b misclassifies its hand strength (e.g., identifying two pair as a full house), it tends to overestimate its chances of winning, leading to aggressive betting and ultimately a complete loss of chips in that hand. This tendency towards overconfidence when it misjudges significantly impacts its overall performance.

These findings underscore the importance of evaluating both intermediate steps and final outcomes when assessing LLM performance. While final scores provide a readily quantifiable measure of success, they can sometimes obscure the underlying reasoning processes and mask strengths or weaknesses in an LLM's strategy, as clearly demonstrated in both the Pong and Texas Hold'em examples. Our introduction of intermediate evaluation provides a crucial perspective, revealing otherwise hidden discrepancies between an LLM's capabilities and its ultimate performance. This granular analysis allows for a more nuanced understanding of LLM behavior, enabling us to identify specific areas for improvement.

### 3.2 Stronger Baselines

This work introduces not only a novel evaluation benchmark for LLMs, but also a set of carefully curated Chain-of-Thought (CoT) prompting strategies specifically designed for the games. We demonstrate that naive approaches, such as simply prompting with "think step by step" are insufficient for eliciting meaningful strategic reasoning, often performing close to random chance. In contrast, our expert-informed CoT prompts, incorporating domain-specific game knowledge, provide robust baselines for future research on strategic reasoning in LLMs (See Figure 3). These curated prompts will facilitate more meaningful comparisons between models and accelerate the

Table 4: LLMs equiped with carefully curated Chain-of-Thought (CoT) prompting clearly outperform Random, while those with generic prompts like "think step by step" perform close to Random.

| Matches | Generic CoT | Curated CoT |
|---|---|---|
| Llama3.1-70b versus Random | 5 wins, 5 loses | 10 wins |
| Gemini 1.5 Flash versus Random | 5 wins, 4 loses, 1 draw | 7 wins, 3 loses |

development of auto-prompting (Zhang et al., 2022) or reflection techniques (Shinn et al., 2024) for complex reasoning of LLMs.

## 4    RELATED WORK

**Benchmarking Reasoning Capabilities**    Various benchmarks aimed at evaluating the core reasoning abilities of LLMs have been developed. Examples include GSM8K (Cobbe et al., 2021) and Math (Hendrycks et al., 2021) for mathematical reasoning, HumanEval (Chen et al., 2021) and MBPP (Austin et al., 2021) for code generation, StrategyQA (Geva et al., 2021) for multi-hop reasoning, and Roberts et al. (2023; 2024b) for geospatial reasoning.

**Addressing Data Contamination**    A major challenge in LLM evaluation is data contamination, where LLMs may have encountered test instances during pre-training, leading to inflated performance estimates (Yang et al., 2023; Xu et al., 2024; Zhang et al., 2024). To mitigate this, research has explored two main avenues: (1) human annotation-based methods, which involve creating new datasets or filtering existing ones to minimize overlap with pre-training data (White et al., 2024; Chiang et al., 2024; Kiela et al., 2021; Zheng et al., 2023; Roberts et al., 2024a); and (2) LLM-driven approaches, which leverage LLMs to generate novel evaluation instances or act as judges (Sprague et al., 2023; Fu et al., 2024; Dubois et al., 2024; Zeng et al., 2023). Both methods have their limitations: human annotation can be resource-intensive, while LLM-driven methods can inherit the biases of the employed LLMs (Panickssery et al., 2024).

**Multi-Agent Evaluation of LLMs in Games**    Recognizing the limitations of single-agent benchmarks (Wu et al., 2023) for assessing LLMs' true capabilities, researchers have turned to multi-agent scenarios, particularly within the context of strategic games. Existing efforts such as GT-Bench (Duan et al., 2024), LLMArena (Chen et al., 2024), and GammaBench (Huang et al., 2024) leverage games like Poker, Hanabi, and other game-theoretic tasks to evaluate LLMs in multi-agent interactions. However, these benchmarks primarily focus on evaluating performance based on game outcomes (e.g., win rate) without considering the correctness of the internal thought chains. Instead, our approach provides more interpretability of the model performance by also evaluating the intermediate results.

## 5    CONCLUSIONS

We introduce GAMEBOT, a novel benchmark for evaluating the capabilities of LLMs at competitive gaming, including 8 diverse games covering a wide range of game types, characteristics and strategies. To be successful, the LLMs must be able to (1) understand the rules of each game, (2) be able to interpret the game state at each turn, (3) provide valid moves, and (4) find a winning strategy – thus our benchmark requires complex reasoning abilities. A key feature of GAMEBOT is the decomposition of the games into 2-3 subproblems targeting specific capabilities. In addition to enhancing the LLMs' decision-making, this enables a fine-grained evaluation of reasoning strengths and weaknesses.

We evaluate 17 frontier LLMs on GAMEBOT and find clear differences in model performance, demonstrating that our benchmark is suitably challenging to differentiate the abilities of the strongest models. Overall, the best-performing models are closed-source, with GPT-4o attaining the highest score. Our refined set of CoT prompts introduces domain expertise and proves to be a much stronger baseline than previous approaches. We hope our benchmark and overall findings help guide research in the important domain of strategic reasoning.

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

## A    LLM VERSIONS

All inference in this work was carried out using API services. Specifically, we used the Vertex AI API (Google, 2024) for models in the Gemini, Claude, Mistral, Jamba and LLaMA 3.1 families, the Reka API (AI, 2024b) for Reka Core and Flash, and the Azure OpenAI service (Microsoft, 2024) for the GPT models.

Here is a list of the specific versions of the models accessed via APIs:

- Gemini-Pro: *gemini-1.0-pro-002*
- Gemini 1.5 Flash: *gemini-1.5-flash-preview-0514*
- Gemini 1.5 Pro: *gemini-1.5-pro-preview-0514*
- GPT-4: *gpt-4-1106*
- GPT-4o mini: *gpt-4o-mini-2024-07-18*
- GPT-4o: *gpt-4o-2024-05-13*
- Reka Flash: *reka-flash-20240904*
- Reka Core: *reka-core-20240415*
- Claude 3 Haiku: *claude-3-haiku@20240307*
- Claude 3 Sonnet: *claude-3-sonnet@20240229*
- Claude 3.5 Sonnet: *claude-3-5-sonnet@20240620*
- Reka Flash: *reka-flash-20240904*
- Reka Core: *reka-core-20240415*
- Jamba 1.5 Large: *jamba-1.5-large*
- Jamba 1.5 Mini: *jamba-1.5-mini*
- Mistral Nemo: *mistral-nemo-2407*
- LLaMA 3.1 {8,70,405b}: *meta/llama3-{8,7,405}b-instruct-maas*

## B    DETAILED PROMPTS

### B.1    OTHELLO

```
You are an expert player of the game Othello. The object of
the game is to have the majority of pieces showing your
colour at the end of the game.

**Game Rules**
1. Othello is played on an 8x8 board, with columns labeled
A-H and rows labeled 1-8.
2. Black pieces: "B"; White pieces: "W".
3. The initial board has black pieces at (D,4) and (E,5), and
white pieces at (D,5) and (E,4).
4. A move consists of "outflanking" your opponent's disc(s),
then flipping the outflanked disc(s)to your colour. To
outflank means to place a disc on the board so that your
opponent's row (or rows) of disc(s) is bordered at each end
by a disc of your colour. (A "row" may be made up of one or
more discs).
5. It can happen that a piece is played so that opponent's
pieces in more than one direction are trapped between your
new piece played and other pieces of yours. In this case, all
the pieces in all viable directions are flipped to your
colour.
```

6. If you have no legal move, your turn is forfeited and your
opponent moves again.
7. The game is over when neither player has a legal move
(i.e. a move that captures at least one opposing piece) or
when the board is full.

**Input**
You will receive a state matrix representing the current game
board:
* Empty space: O
* Black piece: B
* White piece: W
You will also be provided all of the current legal moves. You
are supposed to choose the best move based on your strategic
analysis.

**Output**
Provide your chosen move. Before making a decision,
articulate your internal thinking process. Your performance
will be assessed on both the intermediate thinking results
and the final decision. Follow the thinking process:

1. **Strategic Analysis**
Evaluate every legal move considering factors like:
    (a) Corner Control: It is important to try to occupy the
    four corners of the board, as corner pieces cannot be
    flipped. Output whether you have a move to directly
    occupy the corners. The format is "[Intermediate Thinking
    Results 1: True/False]". Gaining control of the corners
    provides a stable foothold and influences the overall
    position on the board. You should be cautious to occupy
    places exactly next to the corners, as it may lose
    control of the corner easily.
    (b) Edge Control: Edges of the board are less powerful
    than corners but still offer many defensive advantages.
    (c) Piece Stability: It is best to place pieces in stable
    positions to avoid being easily flipped. Stable pieces
    can serve as a foundation for further expansion.
    (d) Frontier: Try to make your pieces which are adjacent
    to empty space (frontiers) less. By doing so, you can
    restrict your opponent's mobility (less choice of moves).
    (e) Wedges: A `wedge' in Othello is when a player can
    place a piece between two of the opponent's stable pieces
    on the edge of the board. This usually occurs when there
    is 1 empty edge space between two pieces of the
    opponent's color, but can occur with any odd number of
    spaces (1, 3 or 5). Wedges are a huge advantage for a
    player who can secure one because they give a strong
    anchor point from which they can eventually win one or
    more corners. If you see an opportunity to create a wedge
    you should almost always take it. They severely limit
    your opponent's viable moves.

```
        For example, if one of the edge is: [(A,1):O (B,1):O
        (C,1):B (D,1):O (E,1):B (F,1):O (G,1):O (H,1):O],
        since (D,1) is an empty edge space between two pieces
        of B, if (D,1) is a legal move for W player, it will
        create a wedge. Output all of the coordinates that
        can create a wedge in the format "[Intermediate
        Thinking Results 2: (X,X), (X,X), ...]".
    (f) Mobility: The number of legal moves available to a
    player. Having more mobility is generally better, as it
    provides more options and flexibility in the game.

Note that capturing large numbers of pieces early in the game
is not always best.

2. **Conclusion**
You should output **Strategic Analysis** before this section.
In this section, based on your previous analysis, clearly
state your decision and reason.

3. **Chosen Move**
    * In this section, only output the chosen move. Do not
    include any other words.
    * The format is: "Chosen Move: (X,X)".
```

## B.2   PONG

```
You are an expert in the Atari Pong game. Your task is to
control the right paddle to defeat the left opponent. Given a
sequence of game frames, your goal is to predict the best
action to win the game. The available actions are defined as
follows: 0 - Stay Still; 1 - Move Up; 2 - Move Down.

Here is some extra information:
Ball Position: The X and Y coordinates of the ball on the
screen.
Your Paddle Position: The Y-coordinate range of the right
paddle. The X-coordinate of the right paddle is always 140.
Opponent's Paddle Position: The Y-coordinate range of the
left paddle. The X-coordinate of the left paddle is always
20.
Y-coordinate of Lower Wall: 16
Y-coordinate of Upper Wall: 176

A larger X-coordinate means relatively right-aligned, a
larger y-coordinate means relatively higher.

Your strategy is that, if the ball is moving towards the
left, simply position your paddle in the middle of the
screen. If the ball is moving towards the right, predict the
trajectory of the ball and adjust your paddle's position to
intercept it. To make a difficult angle for your opponent,
you can intercept the ball near the edge of your paddle.
```

```
Provide your chosen move. Before making a decision,
articulate your internal thinking process. Your performance
will be assessed on both the intermediate thinking results
and the final decision. Follow the thinking process:
**[Observation]** Observe the moving direction of the ball.
Output the moving direction of the ball in the format
"[Intermediate Thinking Results 1: Left Down/ Right Up/ Left
Up/ Right Down]".
**[Thought]** Analyze the trajectory of the ball. Predict the
y-coordinate of the ball when its x-coordinate is the same as
your paddle's x-coordinate. Check until your prediction is
valid, but if you have already check for 3 rounds and it
still invalid, just move your paddle to the middle position.
Compare your current position with the y-coordinate where
your paddle should move to, and decide your movement. Output
the y-coordinate of the ball when its x-coordinate is the
same as your paddle's x-coordinate in the format
"[Intermediate Thinking Results 2: XXX]".
**[Action]** Give the action result based on your thought.

Here is an example.

Input:

Frame 1
{'ball_x': 71, 'ball_y': 136, 'player_x': 140, 'player_y':
[66, 82], 'opponent_x': 20, 'opponent_y': [111, 127],
'upper_bound': 176, 'lower_bound': 16}
Frame 2
{'ball_x': 75, 'ball_y': 144, 'player_x': 140, 'player_y':
[62, 78], 'opponent_x': 20, 'opponent_y': [117, 133],
'upper_bound': 176, 'lower_bound': 16}
Frame 3
{'ball_x': 79, 'ball_y': 152, 'player_x': 140, 'player_y':
[59, 75], 'opponent_x': 20, 'opponent_y': [125, 141],
'upper_bound': 176, 'lower_bound': 16}

Output:

**[Observation]**
Based on the current coordinates of the ball (79, 152), we
observe that the ball's x-coordinate is increasing (79 > 75 >
71), indicating that the ball is moving towards the right.
The ball's y-coordinate is increasing (152 > 144 > 136),
suggesting that the ball is moving upwards.
[Intermediate Thinking Results 1: Right Up]

**[Thought]**
[Calculate the ball's trajectory]
We can calculate the trace using the equation y = mx + b.
Given the points from the last two frames, (x1, y1) = (79,
152), (x2, y2) = (75, 144), we can determine the slope m as
(y2 - y1) / (x2 - x1) = (144 - 152) / (75 - 79) = (-8) / (-4)
= 2. The y-intercept, b, can be found using the equation b =
y1 - mx1 = 152 - 2 * 79 = 152 - 158 = -6. Therefore, the
equation for the trace of the ball is y = 2x - 6. Substitute
x = 140 into the equation, y' = 2 * 140 - 6 = 280 - 6 = 274.
```

```
[Check]
The valid y value lies in the range 16 <= y <= 176. But the
calculated y' 274 > 176, which is invalid. Suggesting the
ball's trace will be changed by rebounding.
[Calculate the ball's trajectory]
From the last observation, the ball is moving upwards, so it
will hit the upper bound and change direction. Now the
updated observation of the ball is moving downwards.
Substituting y = 176 into the equation y = 2x - 6, x = (176 +
6) / 2 = 182 / 2 = 91. After rebounding, the slope m will
change to -2. Using the equation y = -2x + b and substituting
(91, 176), we can find b = y + 2x = 176 + 2 * 91 = 176 + 182
= 358. Therefore, the equation for the trace of the ball is y
= -2x + 358. Substitute x = 140 into the equation, y' = -2 *
140 + 358 = -280 + 358 = 78.
[Check]
The valid y value lies in the range 16 <= y <= 176. Since 16
<= 78 <= 176, it is valid.
[Decision]
The y'-coordinate your paddle should cover is 78. The range
of your paddle's y-coordinate is [59, 75]. 78 is out of the
range of [59, 75], and 75 < 78, suggesting your paddle is
lower than the desired place, so the recommended action
should be 1 - Move Up.
[Intermediate Thinking Results 2: 78]

**[Action]**
1 - Move Up
```

## B.3 SURROUND

```
You are an expert in playing the game Surround in Atari 2600.
Your goal is to survive as long as possible and outmaneuver
your opponent.

**Game Rules**

* The game is played on an 20 x 40 grid, while the edge of
the grid is surrounded by walls.
* You and your opponent leave a trail of walls behind you as
you move.
* Colliding with a wall ends the game.
* You can only move to empty spaces (value 0).

**Goal**

Develop a winning strategy by analyzing the game state,
predicting your opponent's moves, and making intelligent
decisions to survive and trap your opponent. To prolong your
survival, you must carefully plan your path to conserve
space. Furthermore, you should try to surround your opponent
with walls, making them run out of room and be forced to run
into a wall.

**Input**
```

```
You will receive a moving trace recording every position you
have been, and a state matrix representing the current game
board:

* Empty space: 0
* Wall: 1
* {} last position: 2
* {} current position: 3
* {} last position: 4
* {} current position: 5

**Output**
Provide your chosen move. Before making a decision,
articulate your internal thinking process. Your performance
will be assessed on both the intermediate thinking results
and the final decision. Follow the thinking process:

1. **Current Position Analysis**
   * State the coordinates of your current position (row,
   column) with value {}. The top-left corner's coordinates
   are (0, 0).
   * According to the given game state, extract all the
   values adjacent to your current position in 4 directions.
   The format is "[Intermediate Thinking Results 1: Up X,
   Down X, Left X, Right X]", where X is the value at that
   position, but if the position is out of the border, set X
   to be -1.
   * Example: "[Current Position]: (10,15). [Up] (9,15): 1
   (Wall); [Down] (11,15): 0 (Empty Space); [Left] (10,14): 0
   (Empty Space); [Right] (10,16): {} (My last position).
   [Intermediate Thinking Results 1: Up 1, Down 0, Left 0,
   Right {}]."

2. **Valid Actions**
   * List all possible move actions based on the available
   empty spaces around your current position. Output in the
   format [Intermediate Thinking Results 2: X, X, ...], where
   X is the available action. If there are no valid actions,
   output [Intermediate Thinking Results 2: None].
   * Example: "[Intermediate Thinking Results 2: Move Down,
   Move Left]"

3. **Strategic Analysis**
   * Explain your reasoning for choosing the final action,
   considering factors like:
     * Long-term survival: Creating open space for future
     moves. Make sure not to trap yourself given the input
     game state. You should at least ensure 10 continuous
     empty cells for future movement. For every valid action,
     find the empty space and output the result. You can stop
     the process when you already found 10 in total. For
     example, suppose the partial game state is
     (0,23):1  (0,24):1  (0,25):1  (0,26):1  (0,27):1
     (1,23):0  (1,24):{}  (1,25):0  (1,26):1  (1,27):1
     (2,23):0  (2,24):{}  (2,25):0  (2,26):0  (2,27):1
     (3,23):0  (3,24):1  (3,25):0  (3,26):1  (3,27):1
```

```
        (4,23):0   (4,24):1   (4,25):1   (4,26):1   (4,27):1
        For moving right, the position would become (1, 25).
        Continue finding any adjacent cells with 0 in all
        directions for (1, 25).
        1. Found empty: [(1, 25)]
        For (1, 25). Up (0, 25): 1, Right (1, 26): 1, Left (1,
        24): {}, Down (2, 25): 0 (new empty)
        2. Found empty: [(2, 25)]
        For (2, 25). Up (1, 25): 0 (added empty), Right (2, 26):
        0 (new empty), Left (2, 24): {}, Down (3, 25): 0 (new
        empty)
        3. Found empty: [(2, 26), (3, 25)]
        For (2, 26). Up (1, 26): 1, Right (2, 27): 1, Left (2,
        25): 0 (added empty), Down (2, 27): 1
        For (3, 25). Up (2, 25): 0 (added empty), Right (3, 26):
        1, Left (3, 24): 1, Down (4, 25): 1
        4. No more new empty found, end the process. Union of
        the found empty: [(1, 25), (2, 25), (2, 26), (3, 25)],
        total 4 cells, less than 10.
        So we should not move right in this circumstance.

        Note that you should strictly follow the analyzing
        process shown in the example step by step for all valid
        actions. Output whether the valid actions will lead to a
        safe path with at least 10 continuous empty cells for
        future movement. The format is "[Intermediate Thinking
        Results 3: 'Valid Action' Safe/Unsafe, ...]". For
        example, "[Intermediate Thinking Results 3: Move Right
        Unsafe, Move Left Safe]".

        * Trapping the opponent: Forcing them into a smaller
        area.
        * Risk assessment: Avoiding potential collisions with
        walls or getting trapped yourself.

    4. **Conclusion**
        * Based on your previous analysis, clearly state your
        decision and reason.

    5. **Chosen Action**
        * In this section, only output the chosen action. Do not
        include any other words.
        * Example: "Move Left"
```

## B.4 CHECKERS

```
    You are an expert player of the game Checkers. Checkers is a
    classic board game, known as Draughts in England. The
    objective of the game is to capture all the opponent's pieces
    by jumping over them.

    **Game Rules**
```

* Game Basics: Checkers is played on an 8x8 chequered board, with columns and rows both labeled 0-7, alternating between 32 dark and 32 light squares. Each player starts with 12 pieces, placed on the dark squares of the board. Black player's pieces start at row 5-7, and white player's start at row 0-2.
* Game Play:
1. Move Only on Dark Squares: Pieces can only move diagonally on the dark squares, the light squares of the board are never used.
2. Move Only One Square at a Time: A normal move is moving a piece diagonally forward one square toward the opponent. You cannot move onto a square that is occupied by another piece.
3. Capture Pieces With Jumps: A piece making a capturing move (a jump) leaps over one of the opponent's pieces, landing in a straight diagonal line on the other side. Only one piece may be captured in a single jump; however, multiple jumps are allowed during a single turn. When a piece is captured, it is removed from the board.
4. Jumps (or Captures) Must Be Made: If a player is able to make a capture, there is no option; the jump must be made. If more than one capture is available, the player is free to choose whichever he or she prefers.
5. Pieces Become Kings: When a piece reaches the furthest row from the player who controls that piece, it becomes a king. (i.e., Black reaches row 0, White reaches row 7) Kings are limited to moving diagonally but may move both forward and backward. (Remember that normal pieces, i.e. non-kings, are always limited to forward moves.) Kings may combine jumps in several directions, forward and backward, on the same turn. Normal pieces may shift direction diagonally during a multiple capture turn, but must always jump forward (toward the opponent).
6. A player wins the game when the opponent cannot make a move. In most cases, this is because all of the opponent's pieces have been captured, but it could also be because all of their pieces are blocked in. The game ends in a draw if the exact same board state has come up three times.  This is to avoid a situation with two pieces left just moving around never being able to capture each other. The game also ends in a draw if there have been 40 moves (20 for each player) with no piece captured.

**Input**
You will receive a state matrix representing the current game board:
* Empty space: _
* Black normal piece: b
* Black king piece: B
* White normal piece: w
* White king piece: W
Coordinate (a,b) means position at row a and column b (zero-based indexing, starting from row 0 and column 0).

You will also be provided all of the current legal moves. You are supposed to choose the best move based on your strategic analysis.

```
**Output**
Provide your chosen move. Before making a decision,
articulate your internal thinking process. Your performance
will be assessed on both the intermediate thinking results
and the final decision. Follow the thinking process:

1. **Strategic Analysis**
Evaluate every legal move considering all of the listed
factors:
    (a) Center Control: This consists of occupying the center
    by moving your pieces into it and by jumping toward the
    center when you have the option of jumping more than one
    way.
    The central squares are more critical to control than the
    edges. All the squares are important, of course, and
    sometimes a well-placed piece on the side of the board is
    advantageous. Again, don't ignore the position on the
    board. But if you have a choice between moving or jumping
    to the side or to the center, go toward the center.
    Why does this help? Because a centralized piece has more
    options.
    * It has two possible moves, while an edge piece only has
    one.
    * It can reach either side quickly if an opportunity
    arises.
    * It can prevent your opponent from attacking a weakness
    on the opposite side.

        (b) Get a King: It is very beneficial to get King
        pieces since King pieces can also move backward.
        Black should try to reach row 0. White should try to
        reach row 7.
        * Output all of the moves that give you a new king
        piece. The format is "[Intermediate Thinking Results
        1: (X,X)->(X,X), ...]". If no such a move, output
        "[Intermediate Thinking Results 1: None]".

    (c) No worthless die: Example: Consider a game board
    [(0,5):_, (0,3):_, (1,4):w, (2,3):_, (3,2):b]. For White,
    move from (1,4)->(2,3) is a bad move, since it would be
    captured by (3,2):b immediately, but no capture back
    since (0,5) and (0,3) are both empty.
    * Output all of the bad moves that lead to a worthless
    die. The format is "[Intermediate Thinking Results 2:
    (X,X)->(X,X), ...]". If no such a move, output
    "[Intermediate Thinking Results 2: None]".

    (d) Protect Your King Row: Getting the first king is a
    huge advantage among less-skilled players. The natural
    tendency is to refrain from moving your back row. This is
    certainly better than carelessly moving them out without
    any plan. But there's a better way.
```

```
        If you don't move your back four pieces, that leaves
        you eight pieces to advance against your opponent. If
        your opponent does move some of the back pieces, your
        eight could be clashing with ten or twelve pieces.
        This could easily leave you on the wrong side of some
        exchanges.
        The general strategy used by experts is to advance
        two of the four back pieces. This gives you an
        attacking force of ten while leaving enough of a
        defense to seriously slow down any Kinging attempts.
        If you're playing someone who doesn't want to move
        any back row pieces, you'll have the advantage.
        You'll be advancing ten pieces against eight while
        still having your back row sufficiently defended.
        So, which two pieces do you leave behind? If you look
        at the back row, you'll find there's only one pairing
        that successfully defends every square in front of
        them. For black, it's the pieces on (7,2) and (7,6);
        for white, it's the pieces on (0,1) and (0,5). Leave
        those two as long as you reasonably can and bring the
        other two into your attack.

(e) Keep a Strong Formation: Pieces grouped together tend
to be stronger than ones that are separated. Advance your
pieces collectively, using the ones behind to support the
ones in front. For example, if part of the game board is
[(2,3):w, (3,4):w, (4,5):b, (5,6):_] and it is Black's
turn, since (5,6) is empty, (4,5):b faces the danger to
be captured by (3,4):w. Black may consider to move
(4,5):b otherwhere or move other pieces to (5,6) to keep
a strong formation.
        A solid mass of pieces isn't as vulnerable to double
        or triple jumping attacks. It also can't be easily
        broken up. If your opponent forces exchanges with the
        front pieces, you'll still have connected pieces
        behind them to continue your charge.
        Amateurs often exchange pieces randomly just to
        simplify the game. Instead, try to build a strong
        formation. When your opponent feels the pressure and
        starts initiating exchanges, you'll find your
        superior development leaves you in a stronger
        position.

(f) The Two-for-One Shot: This is probably the most basic
tactic available to the checker player. Getting one piece
jumped and jumping two in return feels really great. In
games between novices, these situations just seem to
happen. Really, though, they're not coming out of
nowhere. Knowing how to create these shots will win you a
lot of games.
```

```
        For example, if the game board is [(1,4):w, (2,7):_,
        (3,4):w, (3,6):w, (4,5):_, (5,4):b, (5,6):b, (6,7):b,
        empty else] advancing the black piece (5,6) -> (4,5)
        forces the white piece (3,4) to capture this black
        piece and become (5:6):w. Black loses a piece but but
        now the board turns into [(1,4):w, (2,7):_, (3,4):_,
        (3,6):w, (4,5):_, (5,4):b, (5,6):w, (6,7):b], which
        gives Black a double jump: now (6,7):b can jump over
        (5,6):w to (4,5), and continue to jump over (3,6):w
        to (2,7). So Black sacrifice one piece to capture two
        White's pieces.
        For Three-for-One or Three-for-Two Shot, they work on
        the same principles.
        * Output all of the moves that can create a
        Two-for-One Shot in the format "[Intermediate
        Thinking Results 3: (X,X)->(X,X), ...]". If no such a
        move, output "[Intermediate Thinking Results 3:
        None]".

    (g) Attacking Triangles and Triplicates: A group of three
    connected pieces, either in a triangle or along a
    diagonal, can quickly become a liability if the middle
    piece can be removed. That will leave two spaced pieces
    vulnerable to a double jump.
        Example: Consider a game board [(0,5):w, (1,2):w,
        (2,3):_, (3,2):b, (4,1):b, (4,3):b, (5,0):W, (5,4):_,
        empty else], Black's pieces are in a triangle
        formation, and White has a King on square (5,0).
        White can remove the middle of the triangle by
        advancing (1,2) to (2,3), forcing Black (3,2) to jump
        over (2,3):w to (1,4). So (3,2) is empty now. That
        leaves the Black King a double jump. (5,0):W now can
        jump over (4,1):b to (3,2) and jump over (4,3):b to
        (5,4).

2. **Conclusion**
You should output **Strategic Analysis** before this section.
In this section, based on your previous analysis, clearly
state your decision and reason.

3. **Chosen Move**
    * In this section, only output the chosen move. Do not
    include any other words.
    * The format is: "Chosen Move: (X,X)->(X,X)".
```

B.5 TIC-TAC-TOE

```
You are an expert player of the game Tic Tac Toe.

**Game Rules**
1. Tic Tac Toe is played on a three-by-three grid by two
players, X and O.
2. X plays first, and O plays second. Then players alternate
turns.
```

3. The player who succeeds in placing three of their marks in
a horizontal, vertical, or diagonal row is the winner.
4. If a position has been marked, players cannot place marks
here anymore. If all nine squares are filled and no player
has three in a row, the game is a draw.

**Input**
You will receive a state matrix representing the current game
board:
* Empty space: _
* X player: X
* O player: O
The coordinates are zero-based indexing.

**Definition**
Center - The square in the middle surrounded by all the other
squares: [(1,1)]
Edge - A piece bordering the center: [(0,1)], [(1,0)],
[(1,2)], [(2,1)]
Corner - A piece bordered by two edge squares: [(0,0)],
[(0,2)], [(2,0)], [(2,2)]

**Output**
Provide your chosen move. Before making a decision,
articulate your internal thinking process. Your performance
will be assessed on both the intermediate thinking results
and the final decision. Follow the thinking process:
1. **Observations**
Based on the current game state, provide the following
observations:
    * Where are your pieces located?
    * Where are your opponent's pieces located?
    * For all valid moves, check step by step for all
    horizontal, vertical, or diagonal rows: are there any
    potential winning moves to form 3 in a row for you or for
    your opponent?
    Output all of the winning moves for you in the format
    "[Intermediate Thinking Results 1: (X,X), (X,X), ...]".
    If none, output "[Intermediate Thinking Results 1:
    None]".
    Output all of the winning moves for your opponent in the
    format "[Intermediate Thinking Results 2: (X,X), (X,X),
    ...]". If none, output "[Intermediate Thinking Results 2:
    None]".
    Strictly perform the checking process step by step as
    below for all valid moves.
        For example, suppose you are player O, Current Game
        Board:
        (0,0):_ (0,1):O (0,2):X
        (1,0):X (1,1):O (1,2):X
        (2,0):O (2,1):X (2,2):_
        All legal moves: ['(0,0)', '(2,2)']
        For (2,2), the checking process is:
        Horizontal row: (2,0):O (2,1):X (2,2):?; - (2,0) and
        (2,1) is different, not winning move for O or X
        Vertical row: (0,2):X (1,2):X (2,2):?; - (0,2) and
        (1,2) are both 'X', winning move for X

```
        Diagonal row: (0,0):_ (1,1):O (2,2):?; - (0,0) is
        empty, not winning move for O or X.

        In this example, after checking for all the valid
        moves, the results should be [Intermediate Thinking
        Results 1: None], [Intermediate Thinking Results 2:
        (2,2)].

2. **Strategic Analysis**
From your previous observations, if you have a winning move
after checking, directly choose it. Otherwise if your
opponent have a winning move, block it. If these are not the
case, choose the best move based on the following strategy:
* When playing first (If you are X):
Avoid placing your first piece on an edge square, and keep it
on the center or a corner square. Placing it on an edge
square will leave you vulnerable and give your opponent the
advantage.
1) Center
If you mark the center, your opponent will either place
his/her first piece on an edge or corner piece.
* If they mark an edge, it's incredibly easy to win - There's
no chance of even tying. Simply place your next piece on one
of the two corners furthest from the edge piece. They will
most likely block that move, which in turn gives them an
opportunity to win. Block their move, and suddenly, you have
two ways to win, and your opponent is helpless.
* If they mark a corner, as a smarter opponent would, it's a
little bit more complicated. Place your next mark on the
opposite corner, or the corner that would make a diagonal of
two X's and one O. If they place their next piece on an edge,
they've made a mistake, and you now have two ways of winning,
depending on which edge they placed their O on. Otherwise,
assuming you keep counter-attacking, the game will end in a
tie.
2) Corner
If you play a corner piece first, there are only two
significant response that your opponent can make: Center, or
not center.
* If their first move is away from the center, you should be
able to win. Remember that your first piece is contained in
both a vertical and horizontal row. Your next move should be
in the other corner of the same row you placed your first
piece. They'll likely counter-attack, leaving you an easy
path to victory like placing at other corners to make
connection to two of your previous pieces at a time. This
will work whether they play a corner or an edge piece first
up.
* If their first move is in the center, it's a little bit
trickier. Again, form a diagonal. If their next move is in
the corner, you can trap them by placing your next piece at
the intersection of the row and column of the previous two
X's. If their next move is at an edge, you'll be forced to
settle for a draw.

* When playing second (If you are O):
For your opponent's first move, if it is in
```

```
1) Center
If they choose the center, place your O on the corner
immediately, which will buy you some time. According to the
best strategy, your opponent will place their next X on the
opposite corner to yours. Your next piece should not be
bordering your previous move. Then, it's the simple matter of
continuously blocking and counter-attacking until a tie is
reached.
Even if they don't use this strategy, keep blocking until you
reach a tie.
2) Corner
If they mark a corner, mark the center, or you will almost
certainly lose against a good opponent. Then remember that
there is one outcome in which a tie is possible from above.
Your opponent has two choices, to either form a diagonal or
place their next piece somewhere else. Assuming that their
move forms a diagonal, as the strategy would dictate, stay on
the edges and off the corners. You can force a tie this way.
Else, as usual, keep blocking until a tie is reached.

3. **Conclusion**
In this section, based on your previous analysis, clearly
state your decision for the coordinate to move and your
reason.

4. **Chosen Move**
* In this section, only output the chosen move. Do not
include any other words.
* The format is: "Chosen Move: (a,b)", where a (value 0-2) is
row, and b (value 0-2) is column.
```

## B.6 CONNECT4

```
You are an expert player of the game Connect Four.

**Game Rules**
1. The game is played on a 6x7 grid by two players, X and O.
2. X typically plays first, then players alternate turns to
drop their pieces.
3. The pieces can only be dropped at the lowest available
space within the column.
4. The first player to connect four of their pieces in a row
wins the game.
5. The connection can be horizontal, vertical, or diagonal.

**Input**
You will receive a state matrix representing the current game
board:
* Empty space: _
* Player 1's piece: X
* Player 2's piece: O
The coordinates are zero-based indexing. For example,
"(0,4):X" represents Player 1 has a piece on Row 0, Column 4.
Row 0 is the lowest and Row 5 is the highest.
```

```
**Output**
Provide your chosen move. Before making a decision,
articulate your internal thinking process. Your performance
will be assessed on both the intermediate thinking results
and the final decision. Follow the thinking process:

1. **Observations**
Based on the current game state, provide the following
observations:
* Where are your pieces located?
* Where are your opponent's pieces located?
* Check for all horizontal, vertical, or diagonal lines: are
there any potential winning moves to form 4 in a row for you
or your opponent?
Output all of the winning moves for you in the format
"[Intermediate Thinking Results 1: (X,X), (X,X), ...]". If
none, output "[Intermediate Thinking Results 1: None]".
Output all of the winning moves for your opponent in the
format "[Intermediate Thinking Results 2: (X,X), (X,X),
...]". If none, output "[Intermediate Thinking Results 2:
None]".
Strictly perform the checking process step by step as below
for all valid moves.
For example, assume you are X player and would like to check
for one of the valid move (3,2),
Current Game Board:
    (5,0):_ (5,1):_ (5,2):_ (5,3):O (5,4):_ (5,5):_ (5,6):_
    (4,0):_ (4,1):_ (4,2):_ (4,3):X (4,4):_ (4,5):_ (4,6):_
    (3,0):_ (3,1):O (3,2):_ (3,3):O (3,4):O (3,5):X (3,6):_
    (2,0):_ (2,1):O (2,2):X (2,3):X (2,4):X (2,5):O (2,6):_
    (1,0):X (1,1):X (1,2):X (1,3):O (1,4):X (1,5):O (1,6):_
    (0,0):X (0,1):O (0,2):X (0,3):X (0,4):O (0,5):O (0,6):_

    For (3,2), Check for X:

    - Horizontal: check to left: (3,1):O, not X, stop; check
    to right: (3,3):O, not X, stop. Zero X in total.
    - Vertical: check to down: (2,2):X, (1,2):X, (0,2):X. 3 X
    in total. A winning move for X.
    - Diagonal 1: check to top left: (4,1):_, not X, stop;
    check to down right: (2,3):X, (1,4):X, (0,5):O, stop. 2 X
    in total, not enough.
    - Diagonal 2: check to top right: (4,3):X, (5,4):_; check
    to down left: (2,1):O. 1 X, not enough.

    Check for O:
    - Horizontal: check to left: (3,1):O, (3,0):_; check to
    right: (3,3):O, (3,4):O. 3 O in total. A winning move for
    O.
    - Vertical: check to down: (2,2):X. 0 O in total.
    - Diagonal 1: check to top left: (4,1):_, not O, stop;
    check to down right: (2,3):X. 0 O.
    - Diagonal 2: check to top right: (4,3):X; check to down
    left: (2,1):O, (1,0):X, 1 O, not enough.
```

```
    In this example, after checking for all the valid moves
    besides (3,2), the results should be [Intermediate
    Thinking Results 1: (3,2)], [Intermediate Thinking
    Results 2: (3,2)].

2. **Strategic Analysis**
From your previous observations, if you have a winning move
after checking, directly choose it. Otherwise if your
opponent have a winning move, block it. If these are not the
case, choose the best move based on the following strategy:
    * Look for opportunities to create multiple winning lines
    (for) simultaneously. If you have two discs in a row
    horizontally and two discs in a row diagonally, placing
    your next disc in the right position could lead to a win
    in multiple ways. For example, you have discs at [(0,1),
    (1,2), (2,2), (2,1)], then place your next disc at (2,3)
    would connect two lines: [(0,1), (1,2), (2,3)] and
    [(2,1), (2,2), (2,3)]
    * If your opponent has two consecutive discs in a row
    horizontally, block them from getting a third disc in
    that row. For example, if your opponent has discs at
    [(0,1), (0,2)], then place your next disc at (0,3) or
    (0,0) to block them.
    * Consider the center column as a strategic starting
    point. Placing your disc in the center column can give
    you more opportunities to create winning lines in
    different directions. Make the most of your opening moves
    by playing in the central columns.
    * Plan Ahead: Think one or two moves ahead. Try to
    anticipate where your opponent might be aiming to connect
    their discs and plan your strategy accordingly. For
    example, if your opponent has a winning move on (3,3),
    while (2,3) is not your winning move, you should not take
    (2,3) as your move, avoiding (3,3) to be a valid move for
    your opponent.
    * Try to get your 3 discs in a row with open spaces on
    either end.

3. **Conclusion**
In this section, based on your previous analysis, clearly
state your decision for the position to place your next disc
and give explanation.

4. **Chosen Move**
    * In this section, only output the chosen move. Do not
    include any other words.
    * The format is: "Chosen Move: (a,b)", where a is the row
    number (0-5), and b is the column number (0-6) where you
    want to place your disc.
```

## B.7 TEXAS HOLD'EM

```
You are an expert poker player playing Texas Hold'em.

**Game Rules**
```

1. Texas Hold'em is a popular poker game played with two private cards and five community cards.
2. Both players start with 100 chips to bet, and the player with the most chips at the end of the game wins. If your chips drop to 0, you lose the game.
3. The game consists of four betting rounds: pre-flop, flop, turn, and river. At flop, turn, and river round, three, one, and one community cards are revealed, respectively.
4. At each round, players can choose to Fold, Check and Call, Raise Half Pot, Raise Full Pot, All in.
    – Fold: Discard your hand, forfeiting any potential winning of the pot and not committing any more chips.
    – Check and Call: If no bet has been made, a player can choose to 'Check', which means they do not wish to make a bet, and play passes to the next player. When a player chooses to 'Call', they are committing an amount of chips equal to the previous player's bet or raise to match it.
    – Raise Half Pot: Raise an amount equal to half the size of the current pot.
    – Raise Full Pot: Raise an amount equal to the size of the current pot.
    – All in: Bet all of your remaining chips.
5. The player with the best five-card hand wins the pot.
6. The hands are ranked from highest to lowest: Royal Flush, Straight Flush, Four of a Kind, Full House, Flush, Straight, Three of a Kind, Two Pair, One Pair, High Card.
    Rank 1 – Royal Flush: A, K, Q, J, 10 all of the same suit.
    Rank 2 – Straight Flush: Five consecutive cards of the same suit. Higher top card wins.
    Rank 3 – Four of a Kind: Four cards of the same rank. Higher rank wins; if same, compare fifth card.
    Rank 4 – Full House: Three cards of one rank and two cards of another rank. Higher three-card rank wins; if same, compare the two-card rank.
    Rank 5 – Flush: Five non-consecutive cards of the same suit. Compare the highest card, then the second-highest, and so on.
    Rank 6 – Straight: Five consecutive cards of different suits. Higher top card wins.
    Rank 7 – Three of a Kind: Three cards of the same rank. Higher rank wins.
    Rank 8 – Two Pair: Two cards of one rank and two cards of another rank. Compare the higher pair first, then the lower pair, and then the fifth card.
    Rank 9 – One Pair: Two cards of the same rank. Compare the pair first, then the highest non-paired card, then the second highest, and so on.
    Rank 10 – High Card: If no hand can be formed, the highest card wins. If the highest cards are the same, compare the second highest, and so on. Cards are ranked from A, K, ... to 3, 2, where A is the highest.

**Input**
You will receive the following inputs:
* Your two private cards.
* The revealed community cards.

```
*  Your chips in the pot.
*  Your opponent's chips in the pot.

**Output**
Provide your chosen action. Before making a decision,
articulate your internal thinking process. Your performance
will be assessed on both the intermediate thinking results
and the final decision.
Follow the thinking process:

1. **Strategic Analysis**
Based on your two private cards and the revealed community
cards, evaluate your winning probability.
* At pre-flop: the winning probabilities of given private
hand are listed as below,
[AA:84.9%, KK:82.1%, QQ:79.6%, JJ:77.1%, TT:74.7%, 99:71.7%,
88:68.7%, 77:65.7%, 66:62.7%, 55:59.6%, 44:56.3%, 33:52.9%,
22:49.3%, AKs:66.2%, AKo:64.5%, AK:64.9%, AQ:64.0%, AJ:63.0%,
AT:62.0%, A9:60.0%, A8:58.9%, A7:57.7%, A6:56.4%, A5:56.3%,
A4:55.3%, A3:54.5%, A2:53.6%, KQs:62.4%, KQo:60.5%, KQ:60.9%,
KJ:59.9%, KT:59.0%, K9:57.0%, K8:55.0%, K7:54.0%, K6:52.9%,
K5:51.9%, K4:50.9%, K3:50.3%, K2:49.1%, QJs:59.1%, QJo:57.0%,
QJ:57.4%, QT:56.5%, Q9:54.5%, Q8:52.6%, Q7:50.5%, Q6:49.7%,
Q5:48.6%, Q4:47.7%, Q3:46.8%, Q2:45.9%, JTs:56.2%, JTo:53.8%,
JT:54.4%, J9:52.3%, J8:50.4%, J7:48.4%, J6:46.4%, J5:45.6%,
J4:44.6%, J3:43.8%, J2:42.8%, T9s:52.4%, T9o:49.8%, T9:50.5%,
T8:48.5%, T7:46.5%, T6:44.6%, T5:42.6%, T4:41.8%, T3:40.9%,
T2:40.1%, 98s:48.9%, 98o:46.1%, 98:46.8%, 97:44.8%, 96:42.9%,
95:40.9%, 94:38.9%, 93:38.3%, 92:37.4%, 87s:45.7%, 87o:42.7%,
87:43.4%, 86:41.5%, 85:39.6%, 84:37.5%, 83:35.6%, 82:35.0%,
76s:42.9%, 76o:39.7%, 76:40.4%, 75:38.5%, 74:36.5%, 73:34.6%,
72:32.6%, 72o:31.7%, 65s:40.3%, 65o:37.0%, 65:37.8%,
64:35.9%, 63:34.0%, 62:32.0%, 54s:38.5%, 54o:35.1%, 54:36.0%,
53:34.0%, 52:32.1%, 43s:35.7%, 43o:32.1%, 43:33.0%, 42:31.1%,
32s:33.1%, 32o:29.3%, 32:30.2%]
where XXo means unsuited two cards, and XXs represents two
suited cards. T means 10.
Judge which is your private hand and output the corresponding
winning probability. The format is "[Intermediate Thinking
Results 1: XXX]". For example, if your private hand is
"Diamand 3, Diamand 4", then it is 43s, output [Intermediate
Thinking Results 1: 35.7%].
If the winning probability is larger than 57%, you may
consider to raise or all in. If the winning probability is
less than 43%, you may consider to fold. However, if your
chips and opponent's chips in the pot are the same, you
should consider check before fold. If the winning probability
is between 43% and 57%, you can consider to check and call.

* At flop, turn, and river round, first analyse your best
five-card hand and output your hand ranking according to the
game rules. The format is "[Intermediate Thinking Results 2:
X]", where X is the hands ranking. For example, 3 represents
Rank 3 - Four of a Kind.
```

```
If your hand ranks equal or higher than Rank 8 - Two Pair,
you can consider to raise or all in. If you are rank 10, and
your highest private card is lower than J, you can consider
to fold. Otherwise, you can consider to check and call. If
your chips and opponent's chips in the pot are the same, you
should consider check before fold.

Consider the following factors to determine your next action:
    * Your current hand ranking and the probability of
    improving it.
    * The community cards and potential winning combinations.
    * Your opponents' possible hands and betting patterns.
    * The pot odds and implied odds.
    * Your position at the table and the betting round.
    * You may consider bluff occasionally, but note that it
    is risky and can only be used in a low frequency.

2. **Conclusion**
Based on your previous analysis, clearly state your decision
and reason.

3. **Chosen Action**
    * In this section, only output the chosen action. Do not
    include any other words.
    * The format is: "Fold", "Check and Call", "Raise Half
    Pot", "Raise Full Pot", "All in".
```

## B.8 BARGAINING

```
You are an expert in the game-theoretic bargaining.

**Game Rules**
* The game consists of two players, Player 1 and Player 2.
* In the pool, there are multiple items available for
bargaining. Each item has a different value for each player
(unknown to the other player). But the sum values of the
items are both 30 for each player.
* The players negotiate to share the items. Each player aims
to maximize the total value of items acquired through
negotiation.
* At each round, the player can either accept the opponent's
proposal or propose a new division of the items. If the
proposal is accepted, the game ends, and the players receive
the items according to the proposal. Players are rewarded the
total value of the items they receive.
* After 8 bargaining rounds, the game has 20 percent chance
of ending at each round. If the game ends without an
agreement, both players receive 0 reward.

**Input**
The pool contains 3 items with varying amounts.
You will receive the following inputs:
* A list of the number of each kind of item available for
bargaining.
```

```
* The values of each item for you.
* The bargaining history of the previous rounds.

**Output**
According to the bargaining history, do you agree with the
opponent's latest proposal? If not, provide your proposed
division of the items. Before making a decision, articulate
your internal thinking process. Your performance will be
assessed on both the intermediate thinking results and the
final decision.
Follow the thinking process:

1. **Evaluation of the proposal**
Based on the previous rounds of bargaining, evaluate the
opponent's latest proposal.
* First, calculate the total value of the items for you and
output the result. The format is "[Intermediate Thinking
Results 1: XXX]". For example, if the proposal at last round
is [P1: (3,3,2), P2: (2,1,1)], and you are Player2 with
values of the items [2,5,0], the total value for you is
2*2+5*1+0*1=9. [Intermediate Thinking Results 1: 9].
* Then, make the same calculations for your opponents'
previous proposals. And compare the total values of the items
for you between previous proposals and the latest one. Is
your opponent proposing a better proposal for you?
* According to your opponent's proposals, infer the items
that your opponent values the most.

2. **Strategic Analysis**
Based on your evaluation, make decisions considering the
following factors:
    * Since the total value of the items in pool for you is
    30, if the value you receive is lower than half of the
    sum value, i.e., 15, you should consider to propose a new
    one other than accept it.
    * Consider the round number and the chance of the game
    ending. In the earlier rounds, you may propose a more
    aggressive division to maximize your value, but in the
    later rounds (for example, larger than 8), you may
    consider to be more cooperative to avoid the game ending
    without an agreement.
    * When making new proposals, consider the items that you
    value the most and the items that your opponent values
    the most. If you two have the same most valued item, you
    may consider to propose a division that gives you more of
    that item.
    * Consider the acceptance rate of your proposals. Analyse
    your proposals that are rejected in the previous round
    and make adjustments.
    * You can also consider to hide your valued items at the
    beginning of the game, and reveal them in the later
    rounds to lead your opponent to accept your proposal. But
    note that it is a little bit risky.
    * When making a new proposal, do not make the total value
    of the items less than 15 for you. You can set a higher
    threshold.
```

```
3. **Check Validity**
If you are making a new proposal, check the validity of the
proposal. For example, Pool: [X,Y,Z], Proposal: [P1:
(X1,Y1,Z1), P2: (X2,Y2,Z2)], make sure X1+X2=X, Y1+Y2=Y,
Z1+Z2=Z.
If the proposal is invalid, you need to make a new one.
For the valid proposal, output the total value of the items
for you. Strictly follow the format: "[Intermediate Thinking
Results 2: XXX]".

4. **Conclusion**
In this section, based on your previous analysis, clearly
state your decision and your reason.

5. **Proposal**
   * In this section, only output the proposal. Do not
   include any other words.
   * If you agree with the opponent's proposal, output
   "Proposal: [Agree]". If you do not agree, output your
   proposed division of the items. The format is: "Proposal:
   [P1: (X,X,X), P2: (X,X,X)]", where X is the number of
   items for each kind.
```

## C  LLM OUTPUT SAMPLE

**Gemini 1.5 pro vs GPT-4o in Texas Hold'em**

```
Game state: GPT4o: [Hearts Q, Clubs A] All in
Gemini 1.5 pro: [Spades Q, Clubs 4] ?
Output from Gemini 1.5 pro

1. **Strategic Analysis**
My hand [Spades Q, Clubs 4] is not a strong starting hand in
Texas Hold'em. According to the given table, the winning
probability of Q4o is 47.7%, which is in the medium range.
Considering my opponent has gone all-in pre-flop, it suggests
a very strong hand.

2. **Conclusion**
Given the strength of my opponent's likely hand and my
mediocre starting hand, the risk of losing all my chips is
too high. Folding is the most logical decision.

3. **Chosen Action**
Fold
```

