# OpenReview forum: "GAMEBOT: Gaming Arena for Model Evaluation - Battle of Tactics"
_ICLR.cc/2025/Conference — ICLR 2025 Conference Withdrawn Submission_

### Official Review · Reviewer_UTuz · 2024-11-01

**Soundness:** 2
**Presentation:** 2
**Contribution:** 2
**Rating:** 3
**Confidence:** 5

**Summary:**

The paper introduces GAMEBOT a benchmark for assessing LLMs' reasoning in gaming settings. It tests 17 LLMs across eight diverse games, revealing their strengths and weaknesses in strategic reasoning. The proposed GAMEBOT includes the intermediate subproblems for an in-depth analysis of the LLM decision-making process. This paper provides substantial observations and conclusions for LLMs in competitive gaming scenarios.

**Strengths:**

1. Strategic reasoning in LLMs is interesting and significant. This paper introduces a novel benchmark for the field and provides a new perspective for evaluating LLMs.
2. The experimental setup is thorough and well-designed, involving 8 diverse games that represent various gaming scenarios and testing 17 different LLMs.
3. The paper includes detailed prompt templates to facilitate easy replication of the experiments.

**Weaknesses:**

1. Some contributions in the paper are unclear and overclaimed. For further details, please refer below. The reviewer would like to emphasize that the two main contributions, "**Mitigation of Data Contamination**" and "**Intermediate Thought Evaluation**" are overstated.
- Data contamination typically refers to the leakage of testing data or the overlap between training and testing data during LLM training. The authors’ claim is based solely on dynamic competition; however, when the games have limited state/action space, it remains feasible for LLMs to learn and memorize the game logs. Additionally, the authors did not provide experimental evidence to substantiate their claims. The reviewer suggests that fine-tuning LLMs on the gaming process and comparing their performance to that of standard LLMs would be necessary to confirm whether the proposed GAMEBOT effectively mitigates data contamination.
- In terms of intermediate thought evaluation, the reviewer initially anticipated that this would involve real 1v1 competitions, where intermediate results would be collected and the model's behavior analyzed. However, what the authors termed as intermediate thought evaluation appears to be more of an "**offline**" assessment: generate intermediate or endgame data and ask questions. The authors did not thoroughly analyze how intermediate thought processes occur during an **ongoing** match. Moreover, several details are missing. Please refer below for additional questions.

2. Some experimental settings are unclear to the reviewer, and many critical experimental configurations are not provided (see below).

3. The authors present experimental results but do not draw conclusions about how LLMs behave differently across various gaming scenarios. For instance, it is not discussed whether LLMs perform differently when faced with perfect versus imperfect information or dynamic environments, or whether LLMs are better at spatial understanding compared to solving mathematical equations (as inferred from intermediate results).

4. Details about **Stronger Baselines** are missing. How was it developed, and what assumptions were made? Additionally, the evaluation is limited, involving only two LLMs over 20 matches.

**Questions:**

Question for claimed contributions:
1. The claims below are overly strong and not entirely appropriate, particularly since no surveys or comparative analyses are provided in the paper. The reviewer recommends revising these statements or including comprehensive and specific comparisons that highlight which methods or datasets suffer from issues such as data contamination, lack of legibility, high cost, or inherent biases, along with potential reasons for these shortcomings.

Line 14: "However, existing benchmarks either suffer from data contamination or lack legibility."

Line 50: "These methods can be either costly or susceptible to biases inherent in the employed
LLMs"

2. "Difficulty: The games are challenging enough to differentiate between top-performing models" What is the search/state/action space for each game? For example, how many actions does each game support, and what is the average number of turns/actions/tokens consumed per game? Without these statistics, it is difficult to verify that this benchmark provides challenging scenarios to differentiate between LLMs.

Questions for evaluation:
1. How are the values in each cell of Table 3 calculated? Did the authors conduct matches between each pair of LLMs and then average their "Outcome Evaluation" results? Could the authors also report the number of matches conducted for each LLM/game combination?

Questions for experimental settings:
1. Does this paper support multi-agent competition, such as scenarios where three LLMs play together?
2. Does this paper consider optimal solvers, such as RL/MCTS-based solvers or regret-based solvers for poker games?

Questions for intermediate thought evaluation:
 The authors claim that their intermediate evaluation involves various aspects such as spatial reasoning, collaboration in competition, math equation solving, long-context information extraction, risk management, and pattern recognition. Could the authors provide detailed profiles for each of these categories, along with examples? For instance, in the case of TicTacToe, which subproblems are used to assess spatial understanding, and which ones evaluate long-context information extraction? Additionally, could the authors share the results for each of these categories?

---

### Official Review · Reviewer_2rbm · 2024-11-03

**Soundness:** 1
**Presentation:** 2
**Contribution:** 1
**Rating:** 1
**Confidence:** 5

**Summary:**

To evaluate LLMs’ capabilities beyond superficial pattern recognition. This paper introduces a novel benchmark, so called GAMEBOT, which can evaluate LLMs in competitive gaming environments that addresses these limitations. GAMEBOT decomposes complex reasoning in games into modular subproblems, including rule understanding, game state comprehension, and adherence to strategic instructions. This paper also develops Chain-of-Thought (CoT) prompts that leverage domain knowledge to guide LLMs and automatically validate their intermediate reasoning steps against ground truth, not only the accuracy of final decisions but also the quality of the underlying reasoning process. GAMEBOT contains 17 prominent LLMs across 8 diverse games. GAMEBOT can offer four advantages over existing benchmarks, i.e. Mitigation of Data Contamination, Legibility, Difficulty, Stronger Baselines.

**Strengths:**

1.	GAMEBOT decomposes complex reasoning in games into modular subproblems, including rule understanding, game state comprehension, and adherence to strategic instructions.
2.	This paper also develops Chain-of-Thought (CoT) prompts that leverage domain knowledge to guide LLMs and automatically validate their intermediate reasoning steps against ground truth, not only the accuracy of final decisions but also the quality of the underlying reasoning process.
3.	GAMEBOT has evaluated 17 prominent LLMs across 8 diverse games.

**Weaknesses:**

1.	As a benchmark considering about real-world applications with complex reasoning, games contained in GAMEBOT are still a little simple. Besides, it has a limited range of games. So, it might don’t satisfy the demand of this paper’s purpose.
2.	GAMEBOT has 2 types of evaluation metric. But, the scores from the tables, can not reveal any deep analysis results.
3.	GAMEBOT integrate 17 typical LLMs. There are no interface description, that how users can add their own LLMs or games.

**Questions:**

1.	Fig 1 shows the framework of GAMEBOT. Can the author provides some details form system framework or software design?
2.	Is this benchmark is open-source and released online?
3.	GAMEBOT has decomposed complex reasoning in games into modular subproblems. The authors should provide some clues to read and comprehend from results, such as Tab 2 and 3.

**Details Of Ethics Concerns:**

None.

---

### Official Review · Reviewer_jjWX · 2024-11-03

**Soundness:** 3
**Presentation:** 3
**Contribution:** 2
**Rating:** 5
**Confidence:** 4

**Summary:**

This paper introduces a game environment suite to benchmark reasoning capabilities of large language models through competitive game play. The authors decompose the complex reasoning process of each game into modular subproblems that can be easily evaluated on each LLM. This provides a way to assess performance of a models on intermediate reasoning steps as well as final outcome. The test their benchmark of games on 17 top LLMs and do a qualitative analysis of strengths and weaknesses of these models on corresponding reasoning task.

**Strengths:**

1. The paper decomposes games into subproblems that allow evaluating LLMs not only on final outcomes but also on their reasoning capabilities on the intermediate steps they have taken along the way.

2. The set of 8 games presented in paper capture different aspects higher level reasoning in game playing ranging from spatial recognition to mathematical reasoning to information extraction and pattern recognition.

3. They evaluation provides a deeper insight into behavior of LLMs on different cognitive abilities highlighting their strengths and weaknesses.

**Weaknesses:**

1. The subproblems appear to be hand crafted and not exhaustive.

2. Some games like tic tac toe, connect four have very simple pattern recognition evaluation task which does not align with stated goal of the paper. There could be interesting reasoning questions like asking minimum number of feature steps to win the game or asking to compare between two move and so on.

3. Evaluation metric on intermediate thought evaluation is not very clear.

**Questions:**

1. Can the authors comment on how did they come up these subproblems? Have they used any systematic design principles?

2. How do you compare the LLM generated response to ground truth?

3. Any thoughts on generalizing the benchmarking questions beyond hand crafted ones and to game environments of different sizes?

---

### Official Review · Reviewer_4jeq · 2024-11-05

**Soundness:** 2
**Presentation:** 3
**Contribution:** 2
**Rating:** 5
**Confidence:** 4

**Summary:**

This "GAMEBOT: Gaming Arena for Model Evaluation - Battle of Tactics" introduces a competitive gaming benchmark for evaluating large language models (LLMs. GAMEBOT assesses LLM performance in various games by decomposing complex reasoning into modular subproblems, targeting abilities like rule understanding and strategy execution. The framework leverages CoT prompts for more structured reasoning and includes intermediate evaluations to better gauge the decision-making process. The authors benchmark 17 prominent LLMs across eight diverse games, demonstrating the benchmark's effectiveness in distinguishing model strengths and limitations.

**Strengths:**

1. **Innovative Benchmark for Reasoning Evaluation**: The paper presents a well-designed benchmark that emphasizes reasoning and strategy, addressing limitations in existing benchmarks by evaluating intermediate thinking steps.
2. **Diverse Game Selection**: By incorporating eight games with varying characteristics, including board games, action games, and card games, the benchmark captures a range of reasoning skills like spatial reasoning, collaboration, and risk management.
3. **Clear Evaluation Metrics**: The use of intermediate thought evaluation and final outcome assessment provides a comprehensive measure of LLM capabilities, enhancing the interpretability of model performance.

**Weaknesses:**

1. This paper only includes simple games, and most of the environments have already been covered by "LLMARENA: Assessing Capabilities of Large Language Models in Dynamic Multi-Agent Environments." What is the main difference from LLMARENA?

2. The use of sub-tasks is helpful for fine-grained evaluation, which I find interesting. However, the tasks in this paper are relatively simple, and the sub-task design seems human-crafted, which may limit generalization to more complex, real-world games.

3. Additionally, since the sub-task evaluation results come from the LLM, how is the evaluation accuracy of the LLMs ensured?

4. This paper compares base LLMs like GPT-4o and GPT-4, but what about popular LLM agent algorithms like ReAct, Reflection, or fine-tuning-based methods? These well-structured tasks seem easy to learn via RL.

**Questions:**

1. **Prompt Dependency**: How would the model perform if given generic prompts rather than task-specific CoT prompts? Could the authors discuss the adaptability of prompts across different game contexts?

---

### Author Response · Authors · 2024-12-04
**Response to All Reviewers**

We sincerely appreciate the effort the reviewers dedicated to evaluating our work. After careful consideration, we decided to withdraw the submission. We have thoroughly considered the feedback and will address each problem in the revised manuscript for a future submission.

We summarize the main contributions of our work:

(1) We introduce GAMEBOT, a new benchmark specifically designed for evaluating LLMs in strategic reasoning. GAMEBOT encompasses eight distinct games, carefully selected to assess diverse strategic abilities. Crucially, GAMEBOT is designed for interpretability and presents a challenge to LLMs while inherently mitigating data contamination through its competitive and dynamic setting.

(2) We decompose complex game reasoning into predefined, modular subproblems. Instead of relying on generic "think step-by-step" prompting, we employ detailed, strategically-guided Chain-of-Thought (CoT) prompts infused with domain knowledge. This guides LLMs through each subproblem before action selection. Furthermore, we develop a suite of rule-based algorithms to generate ground truth for these subproblems, enabling rigorous validation of the LLMs' intermediate reasoning steps. This provides crucial interpretability for evaluation.

(3) We evaluate 17 prominent LLMs (GPT, Claude, Gemini, etc) through rigorous one-versus-one competitions (with a random agent baseline), ensuring exposure to diverse game states. We conduct 20 matches for each LLM pair in each game. Our results demonstrate that GAMEBOT presents a challenge to current LLMs, even when provided with detailed CoT prompts.

----

Addressing Reviewer Comments:

Q1. The games and sub-tasks are simple, which does not align with stated goal of the paper.

A: The games are selected to be conceptually straightforward for human understanding, facilitating the use of GAMEBOT by LLM developers. However, it is crucial to note that despite the simple rules, the games pose a non-trival challenge for LLMs. While our curated prompts improve the performance of LLMs, even top-performing models like GPT-4o only achieve an average score of 0.52 (out of 1). The controlled difficulty level is essential for effective evaluation, as overly complex games would render all models ineffective, while excessively simple games would fail to differentiate performance. Furthermore, several games, such as Othello, Checkers, and Texas Hold'em, present considerable strategic depth even for human players.

Q2: Since the sub-task evaluation results come from the LLM, how is the evaluation accuracy of the LLMs ensured?

A: The sub-task evaluation results are not derived from LLM outputs. We have developed deterministic, rule-based algorithms for each subproblem to generate the ground truth, ensuring reliable evaluation of the LLMs' intermediate reasoning steps.

Q3: The intermediate thought evaluation appears to be an "offline" assessment: generate intermediate or endgame data and ask questions.

A: The intermediate thought evaluation is online. It occurs within the context of real 1v1 competitions. Intermediate results are collected, and the model's behavior is analyzed dynamically throughout the ongoing games. This allows us to assess the LLM's reasoning process in a realistic, interactive setting.

Q4: Comparison to related work like LLMARENA and GTBench.

A: Our work differs from existing benchmarks in two key aspects:

**Strategically-Guided CoT Prompting:** We develop curated CoT prompts tailored to elicit effective reasoning in games. We observed that LLMs struggle to generate reasonable actions with generic prompts like "think step-by-step" even for SOT models, hindering performance differentiation. Our detailed CoT prompts lead to more robust results compared to previous work relying solely on generic prompts.

**Interpretability through Intermediate Step Evaluation:** GAMEBOT goes beyond evaluating final game outcomes by rigorously assessing the LLMs' intermediate reasoning steps. We leverage our rule-based algorithms to provide a precise and interpretable measure of the LLM's intermediate thinking results, enhancing the understanding of LLM strengths and weaknesses.

Q5: Details about Stronger Baselines are missing. How was it developed, and what assumptions were made? Additionally, the evaluation is limited, involving only two LLMs over 20 matches.

A: We utilize the same setting as the main benchmarking experiments on TicTacToe. For generic prompts, we use "Think step by step". We will include more experiments and make a more detailed discussion in the next version.

Q6: Evaluation metric is not very clear, and the experiment results need detailed analysis.

A: We will revise the paper to make it clear, and give nuanced analysis in the next version.

Q7: What is the average turn and state space for each game?

A: We will include the details in Table 1.

Q8: Is this benchmark is open-source and released online?

A: The benchmark will be released online and open-sourced.

---

### Note · Authors · 2024-12-04

I have read and agree with the venue's withdrawal policy on behalf of myself and my co-authors.